# Development and Validation of Prognostic Models for Oral Squamous Cell Carcinoma: A Systematic Review and Appraisal of the Literature

**DOI:** 10.3390/cancers13225755

**Published:** 2021-11-17

**Authors:** Diana Russo, Pierluigi Mariani, Vito Carlo Alberto Caponio, Lucio Lo Russo, Luca Fiorillo, Khrystyna Zhurakivska, Lorenzo Lo Muzio, Luigi Laino, Giuseppe Troiano

**Affiliations:** 1Multidisciplinary Department of Medical-Surgical and Dental Specialties, University of Campania “Luigi Vanvitelli”, 80122 Napoli, Italy; dianarusso96@gmail.com (D.R.); marianipier@gmail.com (P.M.); luigi.laino@unicampania.it (L.L.); 2Department of Clinical and Experimental Medicine, University of Foggia, 71122 Foggia, Italy; vitocarlo.caponio@unifg.it (V.C.A.C.); lucio.lorusso@unifg.it (L.L.R.); khrystyna.zhurakivska@unifg.it (K.Z.); lorenzo.lomuzio@unifg.it (L.L.M.); 3Department of Biomedical and Dental Sciences and Morphological and Functional Imaging, Messina University, 98122 Messina, Italy; lfiorillo@unime.it; 4Consorzio Interuniversitario Nazionale per la Bio-Oncologia (C.I.N.B.O.), 66100 Chieti, Italy

**Keywords:** oral squamous cell carcinoma, nomograms, prognostic models, overall survival, prognosis, systematic review

## Abstract

**Simple Summary:**

Prognostic models to choose the right treatment schedule are needed in order to translate into practice a personalized approach. None of these models have been still entered into the clinical practice for what concern oral squamous cell carcinoma (OSCC). In this manuscript we performed a systematic review and subsequent quality assessment of already development prognostic model for OSCC with the aim to take stock of the situation on their possible clinical use.

**Abstract:**

(1) Background: An accurate prediction of cancer survival is very important for counseling, treatment planning, follow-up, and postoperative risk assessment in patients with Oral Squamous Cell Carcinoma (OSCC). There has been an increased interest in the development of clinical prognostic models and nomograms which are their graphic representation. The study aimed to revise the prognostic performance of clinical-pathological prognostic models with internal validation for OSCC. (2) Methods: This systematic review was performed according to the *Cochrane Handbook for Diagnostic Test Accuracy Reviews* chapter on searching, the PRISMA (Preferred Reporting Items for Systematic Reviews and Meta-Analysis) guidelines, and the Critical Appraisal and Data Extraction for Systematic Reviews of Prediction Modelling Studies (CHARMS). (3) Results: Six studies evaluating overall survival in patients with OSCC were identified. All studies performed internal validation, while only four models were externally validated. (4) Conclusions: Based on the results of this systematic review, it is possible to state that it is necessary to carry out internal validation and shrinkage to correct overfitting and provide an adequate performance for optimism. Moreover, calibration, discrimination and nonlinearity of continuous predictors should always be examined. To reduce the risk of bias the study design used should be prospective and imputation techniques should always be applied to handle missing data. In addition, the complete equation of the prognostic model must be reported to allow updating, external validation in a new context and the subsequent evaluation of the impact on health outcomes and on the cost-effectiveness of care.

## 1. Background

Head and Neck Cancer (HNC) is the sixth most common type of cancer across the world with nearly 550,000 new cases per year. Most of HNCs are diagnosed as Oral Squamous Cell Carcinomas (OSCC) and oral cancer ranks eighth among the most common causes of cancer-related deaths worldwide [1,2]. Both pharmacological and surgical protocols for OSCCs diagnosed in early stages are less aggressive and characterized by better outcomes, whilst in advanced stages, very high patients’ morbidity and poor clinical outcomes are expected [3]. Despite the increased knowledge and the encouraging scientific findings of the past 20 years on such diseases, the overall 5-year survival rate for OSCC is still below 50% [4].

Nowadays, the Tumor-Node-Metastasis (TNM) staging system is employed worldwide to predict tumor prognosis and to guide physicians towards the correct treatment choice, however, survival outcomes in patients classified within the same TNM stage class could be dramatically different, with discrepancies in therapy response and tumor management [5].

One of the main limitations of OSCC-related TNM system is its main focus on the anatomical extension of the disease. However, within each staging group, the prognosis can be modified by tumor-related factors, such as genetics, patient age, sex, race or comorbidities. For this reason, the need for a more “personalized” approach to the oncologic patient was underlined in the recent eighth edition of the American Joint Committee On Cancer (AJCC) staging system [6]. It is, therefore, necessary to investigate further prognostic factors to construct prognostic models to carry out a personalized prognosis evaluation [7,8].

Recently, there has been an increased interest in the development of clinical prognostic models and, in particular, in nomograms which are their graphic representation [9]. These are a set of mathematical algorithms that can be used to predict patient outcomes by incorporating multiple variables. Clinic-pathological and genetic variables are mainly incorporated in OSCC prognostic models, showing interesting evidence of their role in patients’ prognosis [10,11]. Purpose of these models is to estimate the probability or individual risk that a given condition, such as recurrence or death, will occur in a specific time by combining information from multiple prognostic factors of an individual [12].

Due to the recent interest in these new prognostic tools, and their potential important role in clinical practice, some guidelines have been defined for explanation and elaboration of clinically useful and correctly elaborated prognostic model. These Guidelines are reported in the Prognosis Research Strategy (PROGRESS) 3 and the Transparent Reporting of a multivariable prediction model for Individual Prognosis or Diagnosis (TRIPOD) [7,13]. In 2016 the AJCC developed the acceptance criteria for inclusion of risk models for individualized prognosis in the practice of precision medicine in the systematic reviews [14]. In the same year, Debray et al. developed a guide for systematic reviews and meta-analyzes of the performance of prognostic models [15]. Additionally, the Prediction Model Risk of Bias Assessment Tool (PROBAST) was also developed to assess the risk of bias and the applicability of diagnostic and prognostic prediction model studies [16].

In this scenario, this study presents a systematic review of clinical-pathological prognostic models with internal validation for OSCC, using the AJCC inclusion criteria and according to current published guidelines.

## 2. Materials and Methods

### 2.1. Protocol

This systematic review was performed according to the *Cochrane Handbook for Diagnostic Test Accuracy Reviews* chapter on searching [17], the PRISMA (Preferred Reporting Items for Systematic Reviews and Meta-Analysis) guidelines [18], and the Critical Appraisal and Data Extraction for Systematic Reviews of Prediction Modelling Studies (CHARMS) [19]. The reviews aim was to evaluate the prognostic performance of nomograms in patients with OSCC. This protocol was designed a priori and registered on the online database PROSPERO (CRD42020219937).

### 2.2. Search Strategy

Studies were identified by using different search engines: Medline/PubMed, ISI Web of Science and SCOPUS. In addition, partial research of the gray literature was carried out through Google Scholar. Furthermore, bibliographies of included studies were handed- revised to find further studies to include in this review. Search operations ended in October 2020. For the search strategy, MeSH terms and free text words were combined through Boolean operators as follow: (prognostic model OR prognostic index OR prediction model OR signature OR risk assessment OR prognostic assessment OR nomogram OR risk score OR model stratification) AND ((OSCC OR “oral cancer” OR tongue) NOT (gastric OR laryngeal OR pharynx OR endocrine OR colorectal OR breast OR prostate OR lung OR salivary OR review OR meta-analysis)).

### 2.3. Eligibility Criteria

To be included, studies had to fulfill the following criteria: (i) characteristics of the prognostic model had to be reported, together with their representative alternative presentation (e.g., scoring system, nomogram, etc.) for patient diagnosed with OSCC undergoing surgery with or without adjuvant therapy; (ii) at least one between with Overall Survival (OS) and Disease-Free Survival (DFS) had to be reported as outcome; (iii) studies had to follow TRIPOD and CHARMS checklist [13,19]; (iv) the prognostic model had to be internally validated; (v) and based on clinicopathological prognostic factors; (vi) that met all the thirteen inclusion criteria described by AJCC [9]; (vii) cohort studies, retrospective studies and studies that performed external validation of a pre-existing model were included; (viii) published in English; (ix) with available full text. We excluded: (i) case reports; case series; reviews and meta-analysis; (ii) studies that intend to modify existing prediction models and not to create new ones; (iii) studies including prognostic models that are not based on measurable markers in resected tumor tissue (saliva, blood, etc.); (iv) studies that met the three AJCC exclusion criteria [9].

### 2.4. Article Selection, Data Collection Process, and Data Items

Articles were independently selected by two of the authors (D.R., P.M.) in multiple steps. First, results of different databases were crossed, and duplicates were electronically removed by EndNote v.X9 software. Subsequently, a manual check was performed to furtherly remove previous undetected duplicates. The first screening for inclusion was performed by reading title and abstract. Full assessment for eligibility was furtherly carried out by full-text reading, judging each study as included, excluded or uncertain, according to the previously listed criteria. A third reviewer (G.T.) acted as an arbiter and calculated a value of k-statistic to ascertain the level of reviewers’ agreement. In cases of disagreement, the same author (G.T.) took a final decision. From each of the selected articles, relevant information were extracted into a data extraction sheet using the TRIPOD and CHAMRS checklist, such as: author, year of publication, country where the study was carried out, the title of the paper, sample size, internal validation sample size, tumor localization sub-site, predictors (candidate and final) used to develop the models, outcome of the model (OS, DFS), method for the internal validation was carried out, modelling method, handling of missing data, model discrimination, model calibration, model presentation, handling of continuous predictors, presence of external validation, type of study.

### 2.5. Risk of Bias Assessment

Risk Of Bias (ROB) within individual studies was assessed by using Prediction model Risk Of Bias Assessment Tool (PROBAST) [16]. PROBAST can be used to assess any type of prognostic prediction model aimed at individualized predictions regardless of the predictors used. The tool comprises four domains—population, predictor, outcome, analysis, questions are answered as “yes”, “probably yes”, “probably no”, “no”, or “no information”. Risk of bias is summarized as “low”, “high”, or “unclear”. The degree of applicability is rated as “low”, “high”, or “unclear” concern. The “unclear” category should be used only when reported information is insufficient. In both cases, for both ROB and applicability, an overall judgment is provided. ROB was assessed separately for development (comprising internal validation) and external validation settings. For articles reporting both model development and external validation, the risk of bias was assessed independently.

## 3. Results

A total of 5972 records were identified in the initial search and were screened by title and abstract by two reviewers. Among these, 66 match our eligibility criteria and were furtherly assessed by full-text reading. At the end of selection process, 6 articles were considered suitable for inclusion in this systematic review [20,21,22,23,24,25]. Details on the selection process and reasons for exclusion are shown on Figure 1.The value of k-statistic resulted 0.87, indicating an excellent level of agreement between reviewers.

### 3.1. Study Characteristics and Model Development

All studies were published between 2014 and 2019. Prognostic models were mainly developed in China (50%, *n* = 3) [22,24,26], the remaining in India (33.3%; *n*= 2) [21,25] and in USA (16.6%; *n* = 1) [23] (Table 1). Patient data were collected retrospectively and hospital-based in four studies [21,23,25,26], while in two studies these were collected from the SEER database [22,24]. Data of patients’ samples and tumor characteristics are summarized on Table 1.

The main investigated prognostic factor was age (100%; *n* = 6) [21,22,23,24,25,26], in four articles T stage [22,23,25,26], N status [22,24,25,26] and sex [21,22,23,26] are inspected, while three studies looked into histological grade [22,24,26] and subsite of the tumor onset [23,24,26]. Main final factors that were found to be independently associated with OS were age and race. Candidates and final prognostic factors included in prognostic models are reported on Table 2. None of the studies evaluated DFS, while OS resulted to be the main outcome (Table 1). Multivariable Cox proportional hazards was used as developer model in 50% of studies [22,24,26], alternatively to a combined modelling method using multivariable Cox proportional hazard regression models and stepdown reduction methods [21,23,25]. Only Montero et al. reported how missing data were handled, by implementation of an imputation technique [22].

In most of the prognostic models (66%, *n* = 4) [22,23,25,26], continuous predictors were dichotomized or categorized, hence the nonlinearity of continuous predictors was assessed. For two prognostic models, cubic splines were used to test for the presence of, a non-linear association between continuous predictors and the predicted outcome [23,26].

All the studies used a nomogram as final presentation [21,22,23,24,25,26]. Methodological characteristics of prognostic models developed are summarized on Table 3.

### 3.2. Validation of the Models

Internal validation was performed in all studies by 1000-time bootstrapping [21,22,23,25,26], except Sun et al. who employed a combined 500-time bootstrapping and 5-fold cross-validation methodology [23].

As a method of discrimination, C-statistics has been used in five studies [21,22,23,24,25]; only one study performed AUC [26].

Four studies reported assessed calibration of the model by means of calibration plots [22,23,24,26], while two did not describe their calibration method [21,25].

In all studies, predictive accuracy was quantified by calculation of the Concordance index (C-index) for each outcome, all the included studies had a C-index higher than 0.6 [21,22,23,24,25,26]. External validation was performed in four studies and C-index was found to be higher than 0.6 in all the articles included [22,24,25,26]. Methodological features of the development and validation of prognostic models are listed on Table 3.

### 3.3. Risk of Bias

PROBAST was used to assess the risk of bias of included studies. Four models presented a low overall bias level [21,22,24,26], while two reported a high overall bias level [23,25]. The overall applicability level resulted to be low in all studies [21,22,23,24,26], except one [25]. Four out of six studies performed external validation of the models [22,24,25,26]. The overall risk of bias was low in three out of four models [22,24,26]. In the external validations, applicability was found to be low in all studies [22,24,25,26]. The risk of bias for each domain of the developed models and the external validations is shown, respectively, on Figure 2 and Figure 3. The applicability for each domain, both for the developed models and for the external validations, is reported in Table 4 and Table 5.

## 4. Discussions

An accurate prediction of cancer survival is very important for counseling, treatment planning, follow-up and postoperative risk assessment in patients with OSCC [27]. Although the use of prognosis models is still relatively new for OSCC, these models are already widely used for other human diseases [28,29,30,31]. It is now well known that cancer-related outcomes are influenced by several factors that are not included in the TNM system. The vast majority of these factors has not been incorporated into the staging system because they may not predict outcome “independently” in multivariate prognosis models, however many of them may work in tandem and have varying degrees of influence on each other [32,33].

This systematic review has yielded a detailed picture of prognostic models for predicting OS in patients with OSCC. Six studies included in this review correctly developed models according to the TRIPOD, all the included studies carried out internal validation of the model and four models were also externally validated [21,22,23,24,25,26]. The majority of models assessed OS in patients with squamous cell carcinoma of the tongue [22,24,26], two assessed all possible sites of tumor onset [21,23], and one model only assessed the buccal mucosa cancer [25]. All models rated OS at five years, except for Bobdey et al [25]. who only rated it at three years; furthermore, Li et al. and Sun et al., also evaluated OS at eight and three years respectively [21,23]. Among the clinical factors, those most included in the models are age, race, martial state, comorbidities and smoking; while among the histopathological ones the most investigated were T stage, N stage and M stage.

This systematic review showed methodological differences in model development. It is well known that the performance of a prognostic model is overestimated when it is just assessed in the patient sample that was used to build the model [34]. Internal validation provides a better estimate of model performance in new patients when done by adjusting overfitting, that is the difference between the accuracy of the apparent prediction and the accuracy of the prediction measured on an independent test set. Resampling techniques are a set of methods to provide an assessment of accuracy for the developed prognostic prediction models [35]. As an exception, Sun et al. [23] used a combined bootstrapping and cross-validation method, although all other studies used 1000-time bootstrapping as a resampling technique. Nevertheless, an evaluation of a model’s performance by using bootstrapping or cross-validation is not enough to overcome overfitting, such type of studies should also apply shrinkage, which is a method used adjust the regression coefficients [36,37]. However, none of the studies used this technique, probably because its usefulness for models with a low number of predictors is unclear [13].

Another important finding from our review is that one-third of the studies did not report on model calibration [38]. Calibration reflects the agreement between the model’s predictions and the observed outcomes. It is preferably reported graphically, usually with a calibration plot [39]. Another key aspect of the characterization of a prognostic model is discrimination, that is, the ability of a forecasting model to differentiate between those who experience the outcome event or not [13]. The most used measure for discrimination is the Concordance Index (C-index), which reflects the probability that for any pair of individuals randomly, one with and one without the outcome, the model assigns a higher probability to the individual with the outcome [40]. For survival models, many c-indices have been proposed, so it is important to underline that, from our results, the most commonly used is the discrimination model proposed by Harrell [41]. In any case, discrimination can vary in a range from 0 to 1 and is considered good when higher than 0.5, considering that all the studies included in this systematic review presented a C-index at least higher than 0.6, all of them showed a good prognostic accuracy [42]. In addition, improvements in study design and analysis are crucial to allow evidence of more reliable prognostic factors that can be incorporated into new prognostic models, or to update existing models, to improve discrimination [43]. Another important finding was the almost total lack of handling of the missing data, except for Montero et al. [22] who carried out the multivariate imputations by chained equations (MICE) [44] before conducting multivariable regression statistical analysis [23]. The absence of a mention of the missing data leads to a so-called “full case analysis”. Including only participants with complete data, as well as being inefficient as it reduces the sample, can also lead to biased results due to a subsample [12]. Additionally, in only two prognostic models, continuous predictors were dichotomized or categorized, and the non-linearity of continuous predictors was examined using restricted cubic splines [23,26].

In the end, only four prognostic models performed external validation, in none of these the population in which the validation was performed was specifically reported and this data also negatively influenced the risk of bias. External validation is preferable to internal validation for testing the transportability of a model since it is impossible for the population, or distribution of predictors, in an independent population to be the same as in the model development population [45]. Secondly, to improve the generalizability of a model, it should ideally be validated in different contexts with different population [46]. Furthermore, in the literature, there are currently no external validation by independent researchers of prognostic models for OS in patients with OSCC. A reliable model should be tested by independent researchers in different contexts to ensure the generalizability of prognostic models [15].

Most of the prognostic models in the literature describe the development of the model, a small number report external validation studies and currently, there are no studies considering clinical impact or utility [7]. Identifying accurate prognostic models and performing impact studies to investigate their influence on decision making, patient outcomes and costs is a fundamental component of stratified medicine because it contributes evidence at multiple stages in translation [47].

Multivariable Cox proportional hazards regression models were used to developing the models, as indicated for survival data [48]. All included prognostic models used nomogram as model presentation, yet none of the prognostic models reported the original mathematical regression formula. This turns out to be highly limiting, firstly because this presentation format is not a simplification of a developed model, but rather a graphical presentation of the original mathematical regression formula, and secondly, because recalibration, and updating of the original formula is necessary to perform validation [49]. Furthermore, it would be advisable to provide readers with the appropriate tools for the interpretation and application of the nomogram [30].

All the studies included in this systematic review had a retrospective design, and therefore showed issues related to missing data and a lack of consistency in predictor and outcome measurement [16]. In addition, both the single-institutional studies and the SEER database lacks critical information. The former, being the cohort of similar patients, may not be relevant in predicting the risk of other patient populations. The second lacks information that could be relevant to prognosis such as comorbidities, chemotherapy and tobacco smoking [50]. Prospective cohort studies should be performed for predictive modeling since they enable not only clear and consistent definitions but also prospective measurement of predictors and outcomes [13,50].

The recognition of the methodological limitations found in the developed models and their external validation were evaluated as a high risk of bias, as indicated in the PROBAST. Domain four (analysis domain) is the one that most influenced the overall risk of bias [16,51].

## 5. Limitations

The main limitations related to this systematic review are due to the very strict inclusion criteria to ensure the high accuracy of the contents. Certainly, having selected only internally validated models and articles written in English has strongly restricted the number of studies included. However, as this is the first systematic review of the literature on prognostic models for OSCC patients, this was done to provide clinicians and researchers with a clear picture of the correct model development method. Future systematic reviews should include a greater number of outcomes (cancer-specific survival, recurrence-free survival, etc.) and include biomolecular prognostic factors in addition to clinicopathological one.

## 6. Conclusions

Based on the findings of this systematic review, the following recommendations could be reported: (i) model development studies should weight for overfitting by carrying out internal validation (by resampling techniques such as bootstrapping) and using shrinkage techniques, (ii) model calibration and discrimination should always be examined, (iii) imputation techniques for missing data handling should always be applied, (iv) non-linearity of continuous predictors should be examined, (v) the complete equation of the prognostic model should always be reported to allow external validation and updating by independent research groups; (vi) prospective studies should be performed to reduce the risk of bias (vii) external validation in a new context and impact assessment on health outcomes and cost effectiveness of care should be carried out.

## Figures and Tables

**Figure 1 cancers-13-05755-f001:**
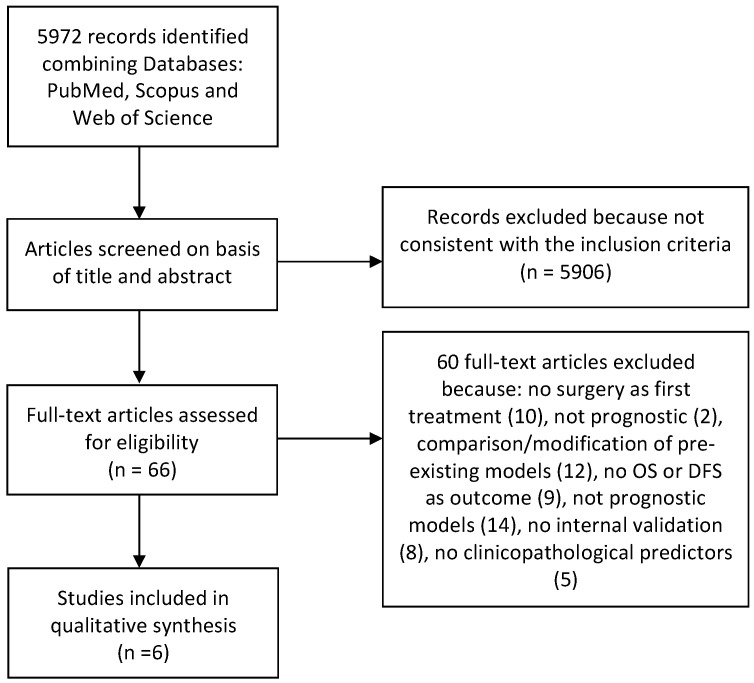
Flow-chart: 5972 records were identified in the initial search and, among them, 66 were further evaluated by reading the full text. At the end of the selection process, 6 articles were considered suitable for inclusion in this systematic review.

**Figure 2 cancers-13-05755-f002:**
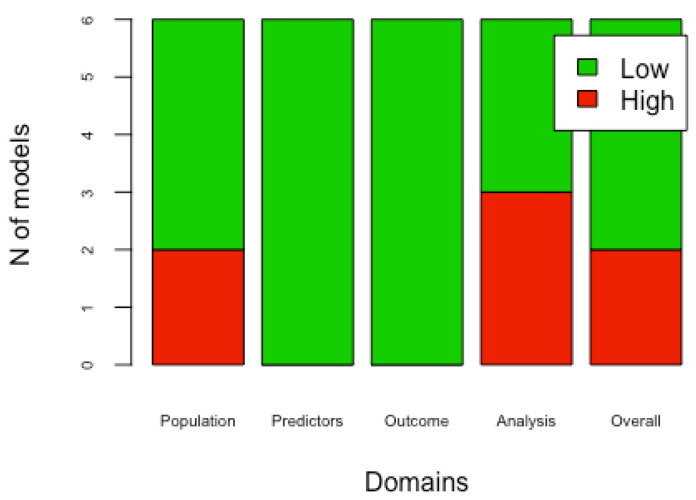
Risk of bias of the developed prognostic models: For each of the six prognostic models included in this systematic review, four domains of bias (population, predictors, outcomes, analysis) were evaluated as “high” or “low”. In this way, the overall risk of bias of each article was assessed.

**Figure 3 cancers-13-05755-f003:**
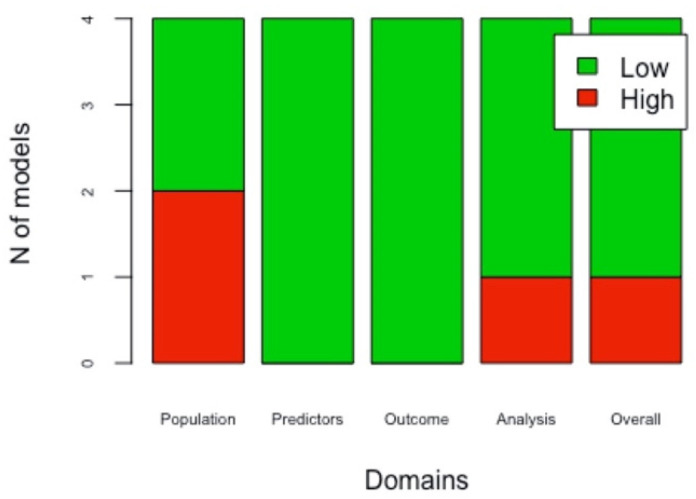
Risk of bias of models’ external validations: For each of the four externally validated prognostic models included in this systematic review, four domains of bias (population, predictors, outcomes, analysis) were assessed as “high” or “low”. In this way, the overall risk of bias of each article was assessed.

**Table 1 cancers-13-05755-t001:** Features of the models included.

Authors	Year	Country	Title	Source of Data	Sample Size	Validation Saple Size	Tumor Site	Outcome	Study
Bobdey [20]	2016	India	Nomogram prediction for survival of patients with oral cavity squamous cell carcinoma	Hospital-based	609	None	Lip, tongue, gum; floor of the mouth; hard palate; cheek mucosa; vestibule of mouth; retromolar trigone	5 years Overall Survival	Retrospective study
Li [21]	2017	China	Nomograms to estimate long-term overall survival and tongue cancer-specific survival of patients with tongue squamous cell carcinoma	Population-based	7587	191	Tongue	5 and 8 years Overall Survival	Retrospective study
Montero [22]	2014	USA	Nomograms for preoperative prediction of prognosis in patients with oral cavity squamous cell carcinoma	Hospital-based	1617	None	Buccal mucosa; tongue; floor of mouth; hard palate; upper gum; lower gum; retromolar trigone	5 years Overall Survival	Retrospective study
Sun [23]	2019	China	Nomograms to predict survival of stage IV tongue squamous cell carcinoma after surgery	Population-based	1085	465	Tongue	3 and 5 years Overall Survival	Retrospective study
Bobdey [24]	2018	India	A Nomogram based prognostic score that is superior to conventional TNM staging in predicting outcome of surgically treated T4 buccal mucosa cancer: Time to think beyond TNM	Hospital-based	205	198	Buccal mucosa	3 years Overall Survival	Retrospective study
Chang [25]	2018	China	“A Prognostic Nomogram Incorporating Depth of Tumor Invasion to Predict Long-term Overall Survival for Tongue Squamous Cell Carcinoma with R0 Resection”	Hospital-based	235	223	Tongue	5 years Overall Survival	Retrospective study

**Table 2 cancers-13-05755-t002:** Predictors included in the prognostic models.

Author Year	Candidate Predictors	Final Predictors
Bobdey 2016 [20]	Age	Age
	Bone infiltration	Clinical lymph node status
	Clinical lymph node status	Comorbidities
	Comorbidities	Differentiation
	Differentiation	Perineural invasion
	Perineural invasion	Stage
	Sex	Tumor thicknesss
	Stage	
	Tumor thicknesss	
Li 2017 [21]	Age	Age
	Grade	Grade
	M stage	M stage
	Martial status	Martial status
	N stage	N stage
	Race	Race
	Radiotherapy	T stage
	Sex	
	T stage	
Montero 2014 [22]	Age	Age
	Alcohol use	Clinical lymph node status
	Clinical lymph node status	Comorbidities
	Comorbidities	Race
	Invasion of other structures	Tobacco use
	Race	Tumor size
	Sex	
	Tobacco use	
	Tumor site	
	Tumor size	
Sun 2019 [23]	Age	Age
	Chemotherapy	M stage
	Grade	Martial status
	M stage	N stage
	Martial status	Race
	N stage	Radiotherapy
	Race	T stage
	Radiotherapy	Tumor site
	T stage	
	Tumor site	
Bobdey 2017 [24]	Age	Bone infiltration
	Bone infiltration	N stage
	Differentiation	Perineural invasion
	Extracapsular spread	
	N stage	
	Perineural invasion	
	Status of surgical margin	
	T stage	
Chang 2018 [25]	Age	Age
	Alcohol use	Depth of invasion
	Body mass index	N stage
	Clinical tumor stage	Neck dissection
	Crossing the midline of the tongue	
	Diabetes	
	Depth of invasion	
	Grade	
	Hypertension	
	M stage	
	Metabolic syndrome	
	N stage	
	Neck dissection	
	Race	
	Sex	
	T stage	
	Tobacco use	
	Treatment	
	Tumor site	

**Table 3 cancers-13-05755-t003:** Methodological characteristics of prognostic models developed.

Authors and Year	Internal Validation	Modelling Method	Handling of Missing Data	Model Discrimination	Model Calibration	Model Presentation	Handling of Continuous Predictors	Non-Linearity	Internal Validation C-Index	External Validation C-Index
Bobdey 2016 [20]	1000-time bootstrapping	Multivariable Cox proportional hazards regression models and stepdown reduction method	n/a	C-statistic	n/a	Nomogram	Mixed: Continuous; Categorical/dichotomous	none	0.7263	none
Li 2017 [21]	1000-time bootstrapping	Multivariable Cox proportional hazards regression models	n/a	C-statistic	Calibration plot	Nomogram	Categorical/dichotomous	n/a	0.709	0.691
Montero 2014 [22]	1000-time bootstrapping	Multivariable Cox proportional hazards regression models and stepdown reduction method	Imputation	C-statistic	Calibration plot	Nomogram	Categorical/dichotomous	Cubic splines	0.67	none
Sun 2019 [23]	Combination of methods: 500-time bootstrapping; 5-fold cross-validation	Multivariable Cox proportional hazards regression models	n/a	C-statistic	Calibration plot	Nomogram	Mixed: Continuous; Categorical/dichotomous	none	0.705	0.664
Bobdey 2017 [24]	1000-time bootstrapping	Multivariable Cox proportional hazards regression models and stepdown reduction method	n/a	C-statistic	n/a	Nomogram	Categorical/dichotomous	n/a	0.7266	0.740
Chang 2018 [25]	1000-time bootstrapping	Multivariable Cox proportional hazards regression models	n/a	AUC	Calibration plot	Nomogram	Categorical/dichotomous	Cubic splines	0.78	0.71

**Table 4 cancers-13-05755-t004:** Applicability of the developed prognostic models.

Author Year	Domain 1	Domain 2	Domain 3	Overall
Bodbey 2016 [20]	Low	Low	Low	Low
Li 2017 [21]	Low	Low	Low	Low
Montero 2014 [22]	Low	Low	Low	Low
Sun 2019 [23]	Low	Low	Low	Low
Bobdey 2017 [24]	Low	Low	High	High
Chang 2018 [25]	Low	Low	Low	Low

For each of the six prognostic models included in this systematic review, four domains (population, predictors, outcomes, analysis) were evaluated as “high” or “low”. In this way, the overall applicability of each article was assessed.

**Table 5 cancers-13-05755-t005:** Applicability of models’ external validations.

PROBAST_External Validation_Applicability				
Author Year	Domain 1	Domain 2	Domain 3	Overall
Li 2017 [21]	Low	Low	Low	Low
Sun 2019 [23]	Low	Low	Low	Low
Bobday 2017 [24]	Low	Low	High	Low
Chang 2018 [25]	Low	Low	Low	Low

For each of the four externally validated prognostic models included in this systematic review, four domains (population, predictors, outcomes, analysis) were assessed as “high” or “low”. In this way, the overall applicability of each article was assessed.

## Data Availability

The data presented in this study are freely available in the article.

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
