# Peer review of "Development and Validation of Prognostic Models for Oral Squamous Cell Carcinoma: A Systematic Review and Appraisal of the Literature"

_cancers, 2021, doi:10.3390/cancers13225755_

Round 1

Reviewer 1 Report

The topic of the manuscript is interesting for sure, oral squamous cell carcinoma is one of the most common malignancy in head & neck pathology with a very poor outcome, especially in advanced stages. The topic is well depicted and exposed, written in correct English except few mistakes, rich of tables that helps the reader. The discussion section is well explained and clear in the description of the results. It is my opinion that the manuscript needs to imrove the bibliography with the following articles that are pertinent for sure with the topic studied.

  • Abbate V, Dell'Aversana Orabona G, Salzano G, Bonavolontà P, Maglitto F, Romano A, Tarabbia F, Turri-Zanoni M, Attanasi F, Di Lauro AE, Iaconetta G, Califano L. Pre-treatment Neutrophil-to-Lymphocyte Ratio as a predictor for occult cervical metastasis in early stage (T1-T2 cN0) squamous cell carcinoma of the oral tongue. Surg Oncol. 2018 Sep;27(3):503-507. doi: 10.1016/j.suronc.2018.06.002. Epub 2018 Jun 2. PMID: 30217309.
  • Sgaramella N, Gu X, Boldrup L, Coates PJ, Fahraeus R, Califano L, Tartaro G, Colella G, Spaak LN, Strom A, Wilms T, Muzio LL, Orabona GD, Santagata M, Loljung L, Rossiello R, Danielsson K, Strindlund K, Lillqvist S, Nylander K. Searching for New Targets and Treatments in the Battle Against Squamous Cell Carcinoma of the Head and Neck, with Specific Focus on Tumours of the Tongue. Curr Top Med Chem. 2018;18(3):214-218. doi: 10.2174/1568026618666180116121624. PMID: 29345578.

Author Response

Reviewer1

We are grateful for the compliments and kind words about our work. We really appreciated your advice and we hope that with them we have managed to improve the quality of this article.

“The topic of the manuscript is interesting for sure, oral squamous cell carcinoma is one of the most common malignancy in head & neck pathology with a very poor outcome, especially in advanced stages. The topic is well depicted and exposed, written in correct English except few mistakes, rich of tables that helps the reader. The discussion section is well explained and clear in the description of the results. It is my opinion that the manuscript needs to improve the bibliography with the following articles that are pertinent for sure with the topic studied.

  • Abbate V, Dell'Aversana Orabona G, Salzano G, Bonavolontà P, Maglitto F, Romano A, Tarabbia F, Turri-Zanoni M, Attanasi F, Di Lauro AE, Iaconetta G, Califano L. Pre-treatment Neutrophil-to-Lymphocyte Ratio as a predictor for occult cervical metastasis in early stage (T1-T2 cN0) squamous cell carcinoma of the oral tongue. Surg Oncol. 2018 Sep;27(3):503-507. doi: 10.1016/j.suronc.2018.06.002. Epub 2018 Jun 2. PMID: 30217309.
  • Sgaramella N, Gu X, Boldrup L, Coates PJ, Fahraeus R, Califano L, Tartaro G, Colella G, Spaak LN, Strom A, Wilms T, Muzio LL, Orabona GD, Santagata M, Loljung L, Rossiello R, Danielsson K, Strindlund K, Lillqvist S, Nylander K. Searching for New Targets and Treatments in the Battle Against Squamous Cell Carcinoma of the Head and Neck, with Specific Focus on Tumours of the Tongue. Curr Top Med Chem. 2018;18(3):214-218. doi: 10.2174/1568026618666180116121624. PMID: 29345578.”

AA: English has been revised. Both suggested articles have been added to the bibliography certainly enriching the quality of our work.

Reviewer 2 Report

The manuscript is interesting for the field of head and neck cancer.

1.Authors should improve the English language and the quality of figures.

2.Why in the search criteria didn’t include 2020 year ?

Please Comment.

I recommend for the publication after minor revision on Cancers.

Author Response

Reviewer2

We really appreciated your advice and we hope that with them we have managed to improve the quality of this article.

“The manuscript is interesting for the field of head and neck cancer.

1.Authors should improve the English language and the quality of figures.”

AA:  English has been revised and figures have been replaced with higher quality images.

“2.Why in the search criteria didn’t include 2020 year ?

Please Comment.

I recommend for the publication after minor revision on Cancers.”

 AA:  Thank you for this comment. The year 2020 was not a priori excluded, there were no restrictions regarding the year of publication in the eligibility criteria. None of the prognostic models developed or externally validated published in 2020 (until October 2020) met our inclusion criteria.

Reviewer 3 Report

In this manuscript, Russo et al. studied that a systematic review of clinical-pathological prognostic models with internal validation for oral squamous cell carcinoma. The main limitations related to this systematic review are due to the very strict inclusion criteria of the contents. In addition, the final results were novelty low.

Major comments:

Only six articles were considered suitable for inclusion in this systematic review, that the conclusions are known to be statistically meaningless.

Author Response

Reviewer3

The manuscript benefited a lot from your revision and we think your criticisms and advice have greatly improved the quality of our work. This is of great importance for our future research work. We hope that the changes made based on your comments will satisfy reviewer' request.

 “In this manuscript, Russo et al. studied that a systematic review of clinical-pathological prognostic models with internal validation for oral squamous cell carcinoma. The main limitations related to this systematic review are due to the very strict inclusion criteria of the contents. In addition, the final results were novelty low.

Major comments:

Only six articles were considered suitable for inclusion in this systematic review, that the conclusions are known to be statistically meaningless.”

AA: We are aware that the number of included studies is not high and that the inclusion criteria used are strict, as we pointed out within the study limits. The eligibility criteria were selected based on what was reported in the Literature on Systematic Reviews of Prognostic Models, in particular, what was reported in "Critical Appraisal and Data Extraction for Systematic Reviews of Prediction Modeling Studies: The CHARMS Checklist" by Moons et al. and “American Joint Committee on Cancer acceptance criteria for inclusion of risk models for individualized prognosis in the practice of precision medicine” by Kattan et al. was strongly taken into consideration. Since this is a systematic review, and therefore with a research protocol designed a priori, it was not possible to change the inclusion criteria in the course of work, in the case we would have included a greater number of outcomes and models that included both clinicopathological and biomolecular factors. Anyhow, our comment prompted us to ask ourselves what is the minimum number of studies to include in a systematic review for it to be considered significant. In the literature, there is no minimum number required for reviews, on the contrary, a minimum of five included studies is recommended for carrying out a meta-analysis. Additionally, we found some interesting work reporting that over 370 reviews with no studies included in the Cochrane Database of Systematic Reviews (Yaffe J, Montgomery P, Hopewell S, Shepard LD. Empty reviews: a description and consideration of Cochrane systematic reviews with no included studies. PLoS One. 2012; 7 (5): e36626. doi: 10.1371 / journal.pone.0036626).

 Regarding content innovation, this is the first Systematic Review on Prognostic Models for OSCC we hope, therefore, it may have enriched the current literature. Following your comments, we have implemented the conclusions to make them more meaningful, emphasizing the impossibility of determining the model with the best prognostic performance due to methodological deficiencies in the included studies. We have also added a section on the future perspectives of studies on this topic. We tried to structure the conclusions to provide readers with "advice" on how to develop and validate future prognostic models on OSCC as our review makes it clear that further and correctly conducted models are needed. It was not our intention to issue a ruling on this topic, on the contrary, ours was intended to be a contribution to a developing field. We sincerely thank you for your review which helped improve our work. We hope that this work if deemed suitable for publication in this journal can be a starting point and a help for other researchers.

Reviewer 4 Report

The need of prognostic models for an individual prognostic calculation in OSCC patients is urgent.

Therefore, the systemtic Review by Russo et al. presents an excellent overview to all research initiatives in this field.

The paper of  Platz H, Fries R, Hudec M: "Computer-aided individual prognoses of squamous cell carcinomas of the lips, oral cavity and oropharynx" , INt J ORal Maxillofac Surg 1992, 21(3): 150-5, PMID 1640127 rweflected this Problem more than 20 years earlier (Discussion, line 227) - and see also Limitations, lines 327-329. A reflection of this paper would enhance the great manuscript. The biometric tests schould be additionally checked by a mathematican.

Author Response

 Reviewer4

We are grateful for your kind words about our work. The suggested article is groundbreaking and emphasizes that the search for an individualized prognosis has deeper roots than current trends.

“The paper of  Platz H, Fries R, Hudec M: "Computer-aided individual prognoses of squamous cell carcinomas of the lips, oral cavity and oropharynx" , INt J ORal Maxillofac Surg 1992, 21(3): 150-5, PMID 1640127 rweflected this Problem more than 20 years earlier (Discussion, line 227) - and see also Limitations, lines 327-329. A reflection of this paper would enhance the great manuscript. “

AA: The article was added to the bibliography allowing us to broaden and improve our discussions.

“The biometric tests schould be additionally checked by a mathematican.”

AA:  Thank you for your comment, we have double-checked the data reported. In addition, data were also verified by a mathematician, however, his help was minimal since no statistical analysis or meta-analysis was carried out. The numerical data was transcribed, to facilitate the reader, from the articles included in the review that were published in well-known and impacted journals and therefore previously subjected to peer review.

Round 2

Reviewer 3 Report

no comment